# End-to-end multimodal structure elucidation from raw spectra combining contrastive learning and evolutionary algorithms

Adrian Mirza [1,2] ✉, Luc Patiny [3] & Kevin Maik Jablonka [2,4,5,6] ✉

Elucidating molecular structures from spectroscopic data remains one of chemistry's most fundamental challenges, typically requiring extensive expert knowledge and manual interpretation of multiple analytical techniques. This is because the structure elucidation problem often has degenerate solutions for a limited set of experimental data. Existing computational approaches are limited to single spectroscopic modalities, require extensive manual pre-processing, and lack the confidence estimates and context necessary for practical application. Here we present SECS, a framework that combines contrastive learning with evolutionary algorithms to automate structure elucidation directly from raw, multimodal spectroscopic data. By aligning embeddings across NMR, infrared, and mass spectrometry, SECS mimics how experts use multiple spectroscopic lenses while providing calibrated confidence scores and relevant database context. On challenging molecular identification tasks, SECS matches expert chemist performance in head-to-head comparisons in a pilot study. The system successfully identifies incorrect structure assignments in published literature and adapts to new chemical domains without retraining by updating its reference database. Our approach demonstrates how synergistic combination of machine learning paradigms can solve analytical bottlenecks that have constrained chemical discovery.

Materials and molecules are typically characterized through spectroscopic and spectrometric measurements, which provide insights into a compound's structure and composition. Scientists need multiple analytical techniques to fully characterize most compounds, as each measurement only provides partial insight into a compound's structure or composition. For instance, infrared (IR) spectroscopy might provide some insights about the presence of functional groups, for which chemists use 1D- and 2D-nuclear magnetic resonance (NMR) experiments to determine their connectivity.

To do so, chemists manually annotate substructures (e.g., [13]C NMR, [1]H NMR, IR, MS) or sublattices (e.g., XRD) in a time-

consuming and error-prone process based on table lookups, intuition, and deductive reasoning[1]. With advances in computational techniques, researchers also employ simulation software to confirm whether the computational predictions for their manually obtained structural model match the experimental evidence[2–6]. For NMR spectroscopy, for instance, Goodman and co-workers pioneered scoring structural candidates using simulations with techniques such as DP-4 and DP-5[7,8].

However, high-fidelity spectral simulations are time-consuming, driving researchers toward machine learning (ML) approaches for predicting spectra from given structures. For instance, there have been

[1]Helmholtz-Zentrum Berlin für Materialien und Energie GmbH, Berlin, Germany. [2]Helmholtz Institute for Polymers in Energy Applications Jena (HIPOLE Jena), Jena, Germany. [3]Zakodium Sárl, Lonay, Switzerland. [4]Laboratory of Organic and Macromolecular Chemistry (IOMC), Friedrich Schiller University Jena, Jena, Germany. [5]Center for Energy and Environmental Chemistry Jena (CEEC Jena), Friedrich Schiller University Jena, Jena, Germany. [6]Jena Center for Soft Matter (JCSM), Friedrich Schiller University Jena, Jena, Germany. ✉e-mail: mail@adrianmirza.com; mail@kjablonka.com

efforts to predict NMR shifts using ML approaches[7,9–14]. However, approaches that only predict spectra given a structure cannot solve the inverse problem of generating structural candidates from measured spectra. They cannot even necessarily be used for unambiguous validation, as a validation would require the enumeration of all plausible compounds and the prediction of their spectra to falsify all alternative hypotheses that could also explain the experimental observations[15].

As a first approximation to the structure elucidation problem, database lookups–also known as dereplication–are frequently used[16–18]. While there has been initial work to build spectral databases and optimized techniques to retrieve structures given an experimental spectrum[19–22], these methods only work on structures for which spectra have been deposited in a database. They cannot be used to predict novel structures.

To address this, ML systems capable of decoding spectra directly into molecular structures have been explored[23–29]. For example, Alberts et al.[24] showed an example of direct "translation" of (peak-picked) $^{13}C$ NMR and $^1H$ NMR to molecular structures. However, these techniques still overlook the reality that chemists must flexibly combine multiple analytical techniques for structure elucidation because a single technique typically only leads to a set of degenerate solutions, as each measurement only provides partial insight into the structure of a compound. Existing multimodal approaches, however, fail to recognize that a system must flexibly combine modalities, as not all of them might be needed or available. For instance, they might require all spectra a model has been trained on as input during inference, inevitably wasting resources for simple systems. They are also not adaptable to easily (without retraining the full system) incorporate new

techniques that might be needed to fully resolve a new set of spectroscopic problems.

As an alternative to fully data-driven approaches, attempts have been made to use substructure-based rules[30]. These systems often require that peak-picked 2D-NMR spectra are provided, as they enable the derivation of bonding patterns.

Very limiting is the fact that most existing computer-assisted structure elucidation (CASE) pipelines[31,32], require manual preprocessing of spectra, such as peak picking, and are, as Devata et al.[29] highlight, very dependent on the quality of this process. Blind studies have found peak-picking–only one aspect of preprocessing–to take on average 84 min per dataset[33–35]. Thus, any approach that relies on preprocessed spectra cannot be used in fully autonomous labs and thus only has a limited impact on accelerating throughput.

In addition, for practical use, such systems must provide relevant context (the most relevant literature examples) and confidence estimates that chemists can trust. Moreover, automated structure elucidation pipelines would prove even more valuable if scientists could easily adjust the knowledge base to new datasets–such as in-house electronic lab notebooks–without retraining models (see Fig. 1).

Current approaches do not meet these requirements.

In this work, we report a system (SECS - structure elucidation from chemical spectra) that produces a ranked list of structural candidates based on flexible combinations of *raw experimental inputs*. It can adapt to new chemical domains and provides relevant context that aids interpretation. It is based on a novel "retrieve, then refine" ansatz that provides a chemically informed, and customizable prior for a generative approach. As we show in more detail in Supplementary Table 1, SECS is, to our knowledge, the only available CASE system that does so.

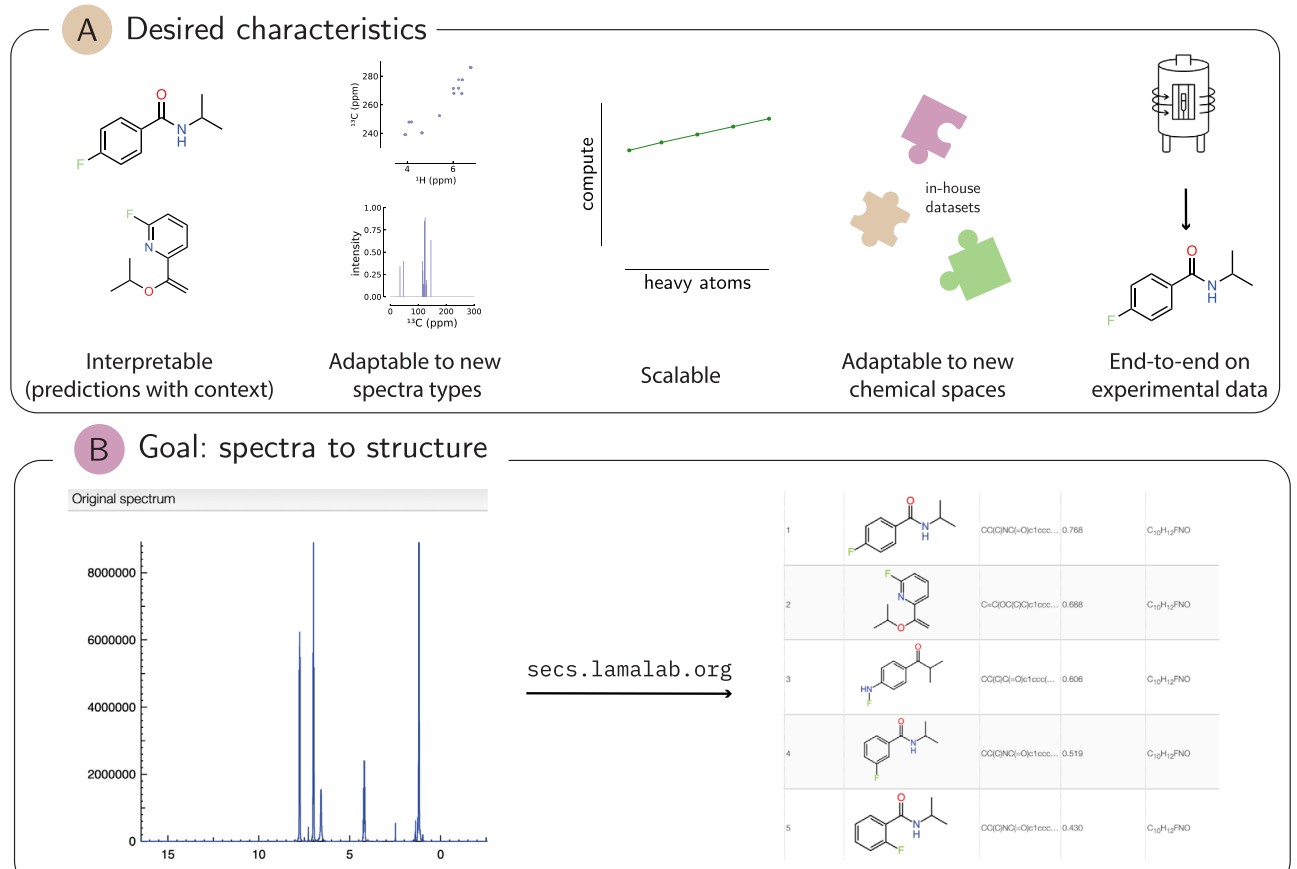

**Fig. 1 | Requirements for computer-assisted structure-elucidation system and our platform. A** We outline five characteristics that would make such a platform most useful. **B** SECS (structure elucidation from chemical spectra) is an automated platform for getting a ranked list of predictions with database context.

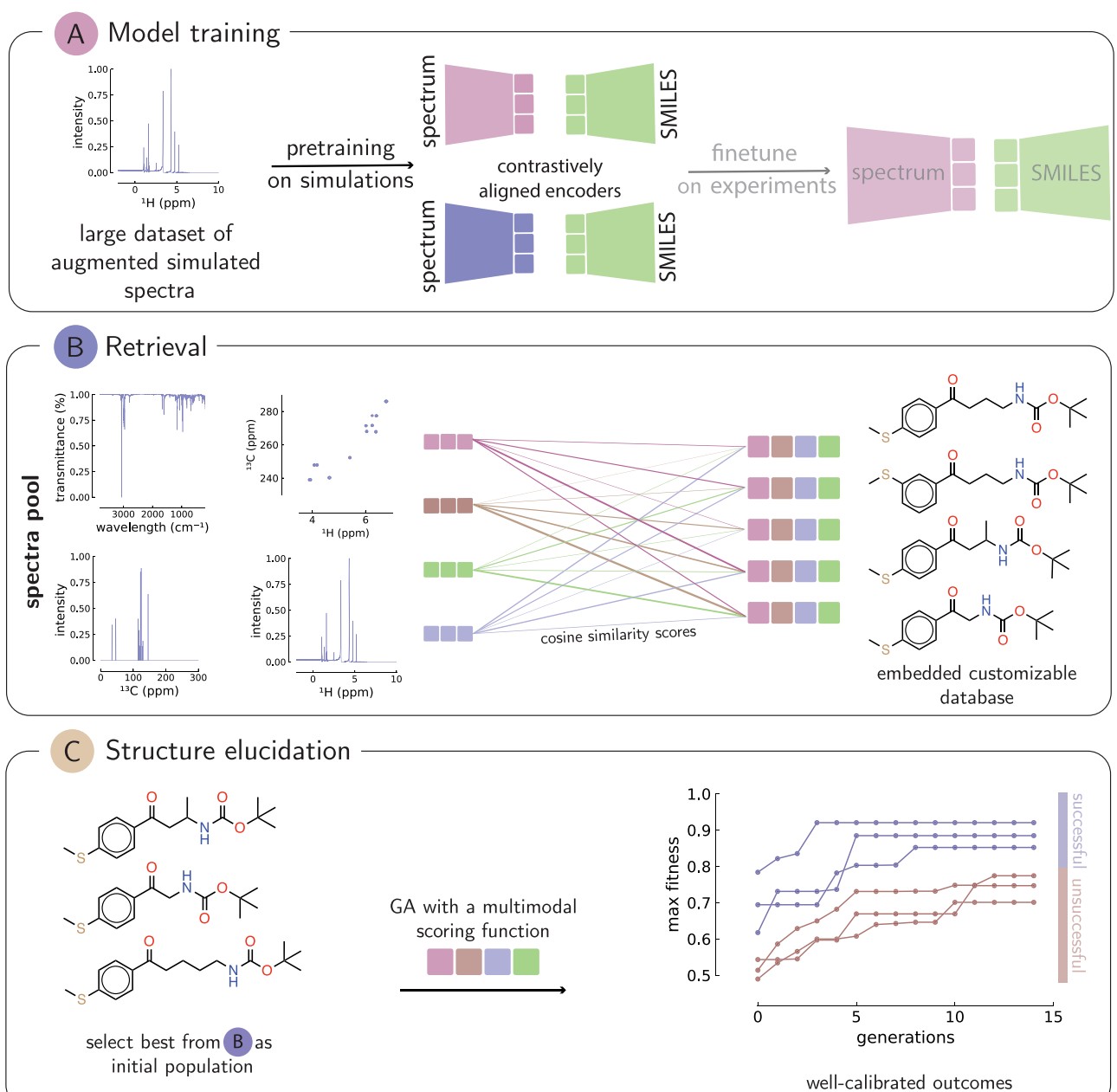

**Fig. 2 | Overview of the SECS workflow. A** The first step is the training of contrastively aligned encoders for various spectra and the SMILES. **B** Toward this goal, contrastively trained models between spectra and molecules are used to query compounds that have similar compositions. This enriches the candidate pool for the starting population. SECS then embeds all the retrieved molecules and ranks the compounds based on a multimodal similarity score between the molecule embeddings and the spectra embeddings. **C** We then run an evolutionary algorithm (GraphGA) on the top-ranked molecules in step (**B**). The output of the ranked list might contain a mixture of novel and retrieved compounds. All compounds and spectra are shown for illustration purposes. GA genetic algorithm, SECS structure elucidation from chemical spectra.

## Results and discussion

We developed a new approach to automated structure elucidation that, for the first time, works with flexible sets of raw experimental inputs.

Current methods compare measured spectra against databases of known spectra. Our method uses contrastive learning to align embeddings of molecules with embeddings of spectra (see Supplementary Fig. 2), effectively tuning encoders to predict the same embedding vector for different measurements or encodings of the same molecule. This alignment enables us to search measured spectra directly against databases of molecules. Searching in databases of molecules using spectra dramatically expands the searchable chemical space, as molecular databases are typically orders of magnitude larger than spectral databases.

The cross-modal retrieval enabled by our approach also solves a key limitation of current approaches—databases cannot be searched across different spectroscopic modalities. Scientists can now retrieve molecules using any combination of available spectra: NMR, IR, MS, or others. As we show below, we can combine this retrieval approach with discrete optimization to systematically refine structure proposals.

To validate our cross-modal retrieval, we tested it on 1000 molecules that we sampled as a test set from a dataset compiled by Alberts et al.[36] based on molecules from patents. We selected only compounds for which we had access to $^1$H NMR, $^{13}$C NMR, IR and HSQC spectra.

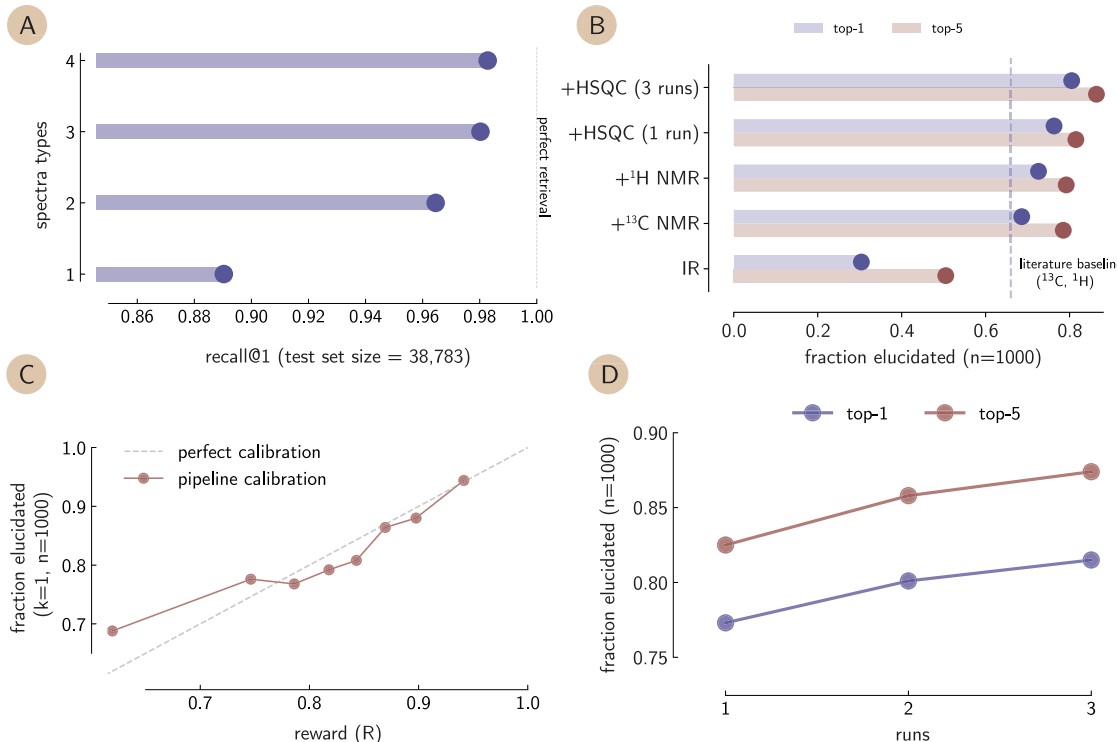

**Fig. 3 | Retrieval performance analysis using combinations of spectroscopic techniques. A** Fraction of molecules correctly retrieved at rank 1 for combinations of spectroscopic techniques, demonstrating improved retrieval accuracy when multiple techniques are combined (follows same order of adding new spectra as panel **B**). **B** Structure elucidation performance as a function of the number of spectra used against the literature baseline from ref. 38 (top-1), indicated by the dashed line. The +indicates the inclusion of a new spectrum type relative to the ones of the bar below. First IR, followed by the addition of $^{13}$C NMR, $^{1}$H NMR (with augmentations), and HSQC. **C** The correlation between the score of the pipeline $R$, demonstrated in Eq. (3) against the average performance of the pipeline. We observe a well-calibrated system. **D** The correlation between the number of runs (using different random seeds and all four spectra types) and the performance, indicating a monotonic increase with the amount of computing time that is spent on the problem.

## Retrieval from databases and the importance of combining spectra

We first analyze how often we retrieve the correct compound by combining various spectroscopic techniques. For this, we encode spectra using our aligned encoders and then calculate the cosine similarities of the encoded spectra against a set of molecule embeddings. Figure 3A shows statistics on how many times we retrieve the correct molecule. We can observe that the combination of multiple spectroscopic techniques enabled by our approach is necessary to achieve a high retrieval performance (shown in Fig. 3A). While we can retrieve around 89% of the compounds when utilizing only one spectrum, the performance increases to 96.4% when we combine two techniques and 98.1% when we combine three techniques and 98.4% when we combine four techniques. That is, the combination of modalities allows us to solve problems that none of the modalities could solve alone (see reduction of extreme ranks in Supplementary Fig. 11 with increasing number of modalities).

While our retrieval approach achieves high performance and benefits from combining multiple spectroscopic techniques, synthetic chemists focus on generating novel compounds that, by definition, do not exist in any database. Retrieval approaches alone cannot identify these new structures.

## Structure elucidation with SECS

For this reason, we extend our system to automatically generate structures that optimally match the available experimental evidence. In practice, chemists might input a set of measured spectra to our pipeline and receive a ranked list of compounds that best match the provided spectra. To do so, we utilize a modified genetic algorithm that directly operates on the structure graph of the chemical structure[37] and evolves the structure (by adding, removing, or changing atoms, bonds, or entire substructures) to better match the experimental evidence. We again use the cosine similarities between the aligned embedding vectors to measure the match between the structures and the spectra (see Eq. (3)).

To make this evolutionary search more efficient, we bootstrap it using an initial pool of molecules created by finding the $N$ molecules whose embeddings are most similar to the embeddings of the input spectra. We look for those compounds within a pool of molecules with similar compositions (where the molecular formula can routinely be obtained from high-resolution mass spectrometry, see Supplementary Note K). In our experiments and web app, this pool comprises PubChem subsets. This retrieval step can be thought of as providing a prior for the generative approach (the genetic algorithm) that refines it retrieved list to better match the experimental evidence.

Finally, our system outputs ranked lists of structure proposals, sorted by similarity between aligned molecular and spectral encodings. These lists contain both compounds from the grounding database and novel compounds proposed by the genetic algorithm. Retrieved compounds provide relevant context, and since we can easily modify compound distributions by changing the database—without retraining models—scientists can straightforwardly optimize the performance of SECS on in-house datasets.

Figure 3B shows how frequently the correct structure is at the very top of this list (top-1) or within the first five entries (top-5). The fraction of times we find the correct structure increases, as expected, if we consider more elements in our ranked list. Interestingly, our system clearly outperforms the current leading approach (which leverages peak lists instead of raw spectra)[38] by around 22%, providing the

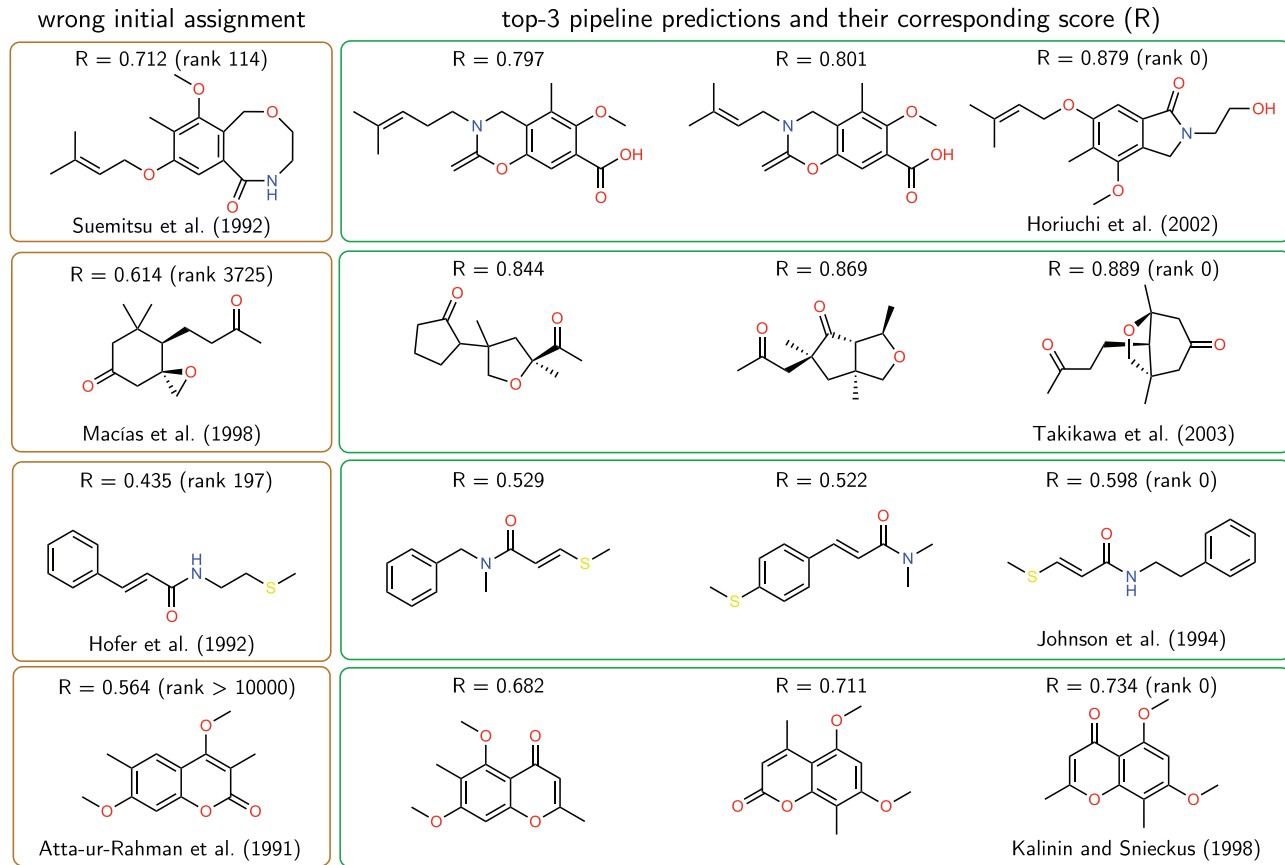

**Fig. 4 | Correcting structure assignments with SECS.** The beige boxes indicate the initially incorrect structures with their respective reward, *R*, according to Eq. (3) (using only $^1$H NMR and $^{13}$C NMR). The green box indicates the top-3 proposed molecules according to our pipeline. To obtain them, we input the simulated spectra and the composition of the correct compounds in our pipeline and then retrieve a starting population, after which the GA attempts to maximize the similarity between the embeddings of the molecules and the spectra[61–68]. Suemitsu et al.[66] identified porritoxin with four analytical techniques. Horiuchi et al.[65] later proposed a new structure via 2D NMR experiments. The NMR analyzed by Macías et al.[68] was later re-evaluated by Takikawa et al.[67]. Greger et al.[63] make use of four different analytical techniques. Johnson et al.[62] later proved via chemical synthesis that the original assignment was incorrect. Atta-ur-Rahman et al.[61] identified an unnamed coumarin that Kalinin and Snieckus[64] later disproved by chemical synthesis. GA genetic algorithm, SECS structure elucidation from chemical spectra.

correct structure in 82% of the cases. At the same time, we also observe that adding spectra after the inclusion of $^{13}$C NMR, does not result in a major boost in performance. This indicates that while many cases can be solved by SECS using only 1D-NMR, there are some problems where the degeneracy can only be resolved by adding additional modalities.

While this high performance makes SECS a powerful tool, it can only find practical utility if it can also indicate the reliability of a prediction. For this reason, we analyzed whether the empirical error rate is correlated with the distance within the latent space (which can be thought of as a so-called conformity score)[39,40]. In our case, we analyze how the multimodal similarity score is related to the accuracy shown in Fig. 3B. The resulting calibration curve is shown in Fig. 3C, where we observe a linear trend between the scaled scores and the proportion of correctly predicted structures, demonstrating a well-calibrated scoring function. This implies that the cosine similarities can be used as reliable confidence estimates. In an autonomous setting, one could thus automatically select the top-ranked candidate. However, experts can often benefit from the additional context provided by returning more than one candidate, which is why we typically return 20 candidates in the SECS application.

### Identifying incorrect structure assignments
Figure 3C shows that if, for example, the reward, *R*, is above a min-max scaled threshold of 0.94, there is a probability of around 94% that the prediction made by the genetic algorithm is correct for examples that are drawn from the same distribution as the ones of the evaluation data. This observation allows us to convert SECS into a practical error detection tool that could be applied upon the entry of spectra in databases. For example, to detect if a spectrum has, by mistake, been attached to the wrong compound in an electronic lab notebook. In those incorrect assignments, we would expect a low similarity score and that our GA can propose a better matching compound.

As realistic case studies, we analyzed compounds for which there was a mistake in the original assignment[41,42]. While many causes might have led to those mistakes, they allow us to test on realistic examples if SECS could have predicted the correct structure if provided with $^1$H NMR and $^{13}$C NMR spectra of this compound. Figure 4 shows four such examples. For these cases, we first compare how close the simulated spectra of the correct compound are to the initially published, but incorrect, structural proposal (beige boxes in Fig. 4). In all cases, SECS indicates a low similarity of the original, incorrect assignment to the spectra of the correct compound. This indicates that SECS potentially could have prevented an incorrect assignment.

However, our SECS pipeline can not only highlight a potentially incorrect assignment, but also propose a potentially better one. For this, we used the entire SECS pipeline (retrieval and GA) to autonomously find a better match to the spectra of the correct compounds (green boxes in Fig. 4). In all cases shown in Fig. 4, SECS identifies the correct structure, with higher scores than those observed for the original incorrect assignment.

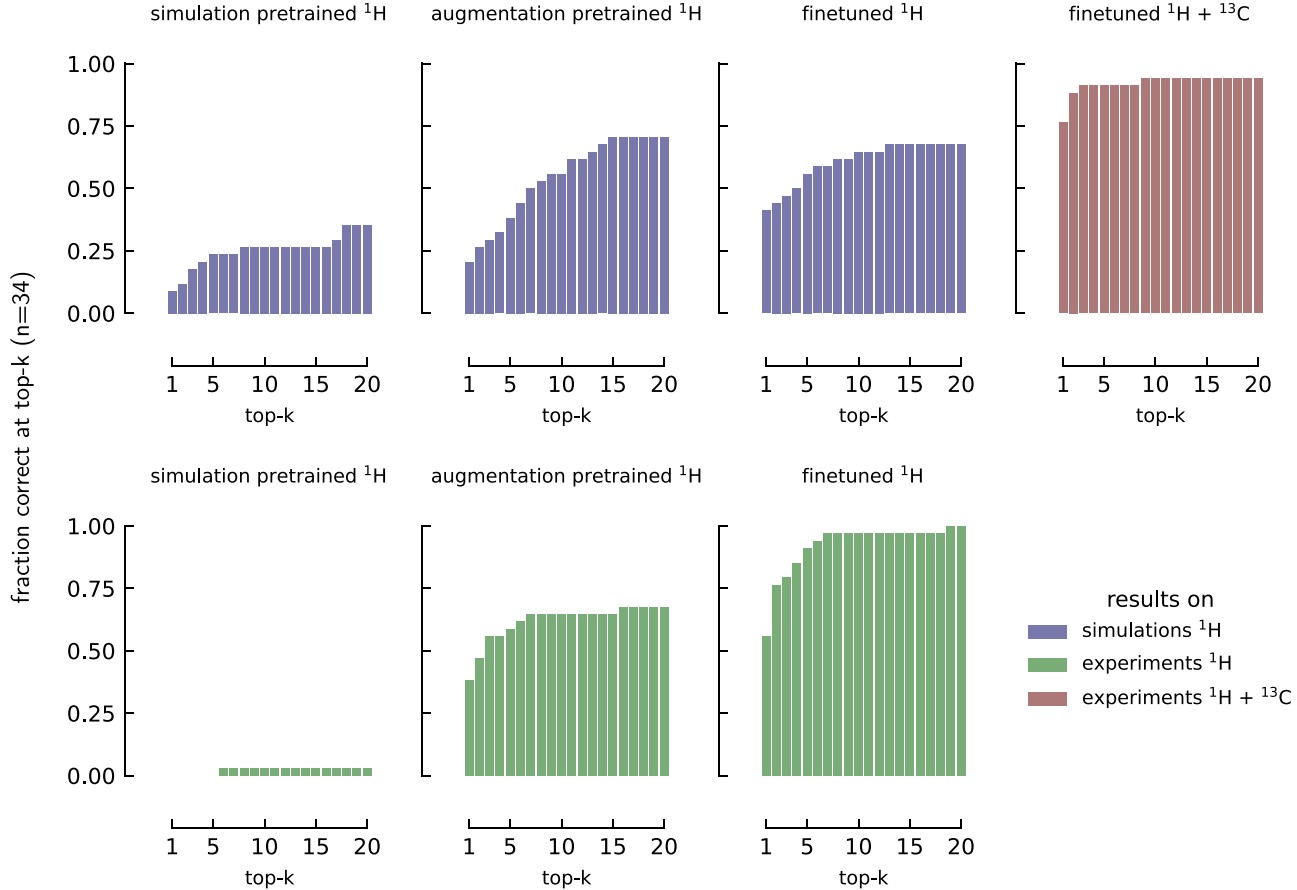

**Fig. 5 | The performance of SECS on ¹H NMR, ¹³C NMR and a combination of the two spectral methodologies.** Here we present the result for 34 molecules for which we had all three spectra: simulated ¹H NMR, experimental ¹H NMR, and experimental ¹³C NMR. The title of each subplot indicates the type of data each model has been trained on, with the legend indicating for what spectra types the results are shown. For example, simulation-pretrained and green indicate that the underlying model has been pretrained on simulated spectra but tested on experimental spectra. "Finetuned" indicates that the augmentation-based model has been trained on an additional experimental dataset. SECS structure elucidation from chemical spectra.

Of course, our approach would not be able to catch all possible mistakes. For instance, our current implementation does not fully handle stereochemistry, but one might easily be extended for this functionality by also including encoders for additional spectroscopic techniques, such as nuclear Overhauser effect spectroscopy (NOESY)[43].

Systems like SECS might also be used for advanced problems like protein structure elucidation, which involves different spectroscopic signatures (e.g., NOESY), databases (e.g., PDB), and validation approaches (e.g., Ramachandran analysis). Future work is also necessary to elucidate mixtures, and might be implemented in future work with, for example, non-negative matrix factorization (NMF)[44] or iterative application of the algorithm[45].

### SECS on experimental data
To further assess the performance of SECS, we tested its performance in multiple experimental case studies.

**In-house dataset.** As a first case study, we compiled an in-house dataset of 34 molecules with experimental ¹H NMR and ¹³C NMR, for which the molecules are not in our training set.

Figure 5 shows the performance of different versions of our pipeline. In the top row, we show the performance on simulated spectra and in the bottom row on experimental spectra. While we find decent retrieval performance with our model trained on the simulation data from ref. 36, we do not find any meaningful results using the experimental inputs.

However, applying data augmentation as described in Fig. 5 leads to a leap in performance on both the simulated and experimental datasets.

By further finetuning the model on a disjoint training set of 2.37k molecules along with experimental spectra generated in-house, we could further improve the performance on experimental data.

We observe that the impact of the augmentation is major, with a performance jump on experimental data from 0% (top-1) for purely simulated spectra to 38.2% for the model trained on augmented spectra, to finally 55.8% at top-1 for the model finetuned on experimental data. We also show that by adding ¹³C NMR (from NMRshiftDB[21,46]) the performance can increase to 88.2% top-1 and 97.1% top-3 performance.

**Recently published NMR spectra.** As a challenging real-world case study, we retrieved spectra that have recently been published in the Chemotion repository[47,48]. None of the spectra appeared in our training sets and, as Supplementary Fig. 14 shows, they expand the chemical space we test (see also molecular weight distribution in Supplementary Fig. 15). We find that SECS also shows a high performance on this dataset, correctly predicting 27.9% for all compounds matching the formula $C_{3-15}H_{4-42}O_{0-5}N_{0-5}Cl_{0-5}$ containing 15 or less heavy atoms and returning the correct compound within the first 20 predictions in 58.5% of the cases (323 compounds). When we consider the entire dataset (1486 compounds) the top-1 performance is 10.3% and the top-20 performance is 24.8%.

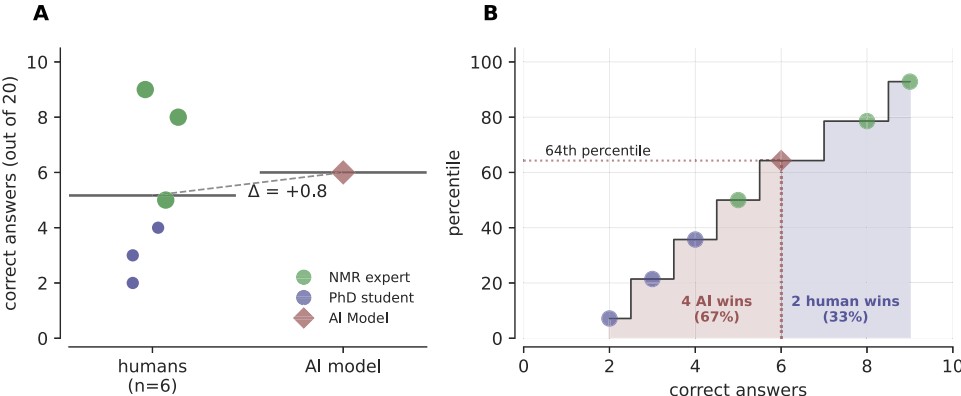

**Fig. 6 | SECS demonstrates expert-level performance in molecular structure elucidation from NMR spectra. A** Individual performance scores for human chemists (circles, $n = 6$) and SECS (brown diamond) on 20 molecular formula identification challenges. Horizontal lines indicate group means. SECS performs comparably to humans with a $\Delta = +0.8$ correct answers (on an equivalence Two One-Sided Tests with a margin of 20% we obtain $p_{TOST} = 0.0287$ indicating that within this margin the human performance and SECS performance do not significantly differ). **B** Cumulative performance distribution. SECS achieves 64th percentile performance among all participants.

## Comparison with human performance

To contextualize SECS performance against human expertise, we conducted a controlled study comparing our tool with experienced chemists on molecular structure elucidation tasks (see Supplementary Note N for details). We selected 20 challenging spectra from the Chemotion dataset at random (within the molecular formula limits stipulated above) and developed a custom web application where participants analyzed $^1$H NMR spectra. We tasked participants with drawing the chemical structures of unknown compounds using molecular editing tools. For each compound, we provided the correct molecular formula and double-bond equivalents and provided immediate feedback on each proposed structure regarding its consistency with these parameters. All participants could use additional resources of their choosing, including databases such as NMRshiftDB. The challenge proved formidable even for experts: the top-performing participant—a scientist specializing in NMR spectroscopy for over two decades—solved fewer than half of the problems, with participants averaging 13.2h to complete all 20 tasks.

Figure 6 demonstrates that SECS achieves expert-level performance on these challenging molecular identification tasks. In head-to-head comparisons (in total, 120 human task attempts), SECS performs comparably to participating chemists. While, due to the time required for experts to solve the tasks, our sample size is limited, these results indicate that SECS not only automates structure elucidation but does so at a level competitive with the domain experts in our study. To enable further validation by the community, we openly release our system and benchmark datasets.

## Discussion

Structure elucidation is at the heart of the chemical sciences. For simple compounds, we can rely on spectral databases and pattern recognition. Computational methods are successful in validating proposed structures by predicting their spectra. For well-characterized systems, we can even use sophisticated simulation software to check whether predictions match experimental evidence. However, these approaches are limited when dealing with complex molecules requiring integration of multiple spectroscopic techniques, or when working with the raw, noisy outputs of real instruments rather than idealized data. More importantly, they cannot propose novel structures that do not exist in existing databases. In addition, existing approaches do not overcome automation bottlenecks as they still rely on manual preprocessing of spectra. Yet chemists have provided vast amounts of paired spectral and structural data across diverse chemical spaces and multiple analytical modalities. Here, we have shown that by aligning multi-modal embeddings and combining them with discrete optimization, this collective knowledge can be converted into a surprisingly powerful tool that works directly with raw experimental spectra. SECS not only matches average human performance, but also provides the contextual information and confidence estimates that chemists need to trust and interpret its predictions.

Our work highlights the impact of synergistically combining machine learning paradigms on problems in chemistry and materials science—particularly for challenges where no single reliable approach exists, but where the community has accumulated substantial experimental knowledge. By learning to bridge the gap between spectral fingerprints and molecular structures, such approaches can transform analytical bottlenecks into computational opportunities, ultimately accelerating the pace of chemical discovery.

## Methods
### Data acquisition
**Spectral data.** Simulated $^{13}$C NMR, $^1$H NMR, IR and HSQC spectra have been taken from ref. 36. They provide a dataset of 794k SMILES-spectra pairs across a variety of techniques (infrared, mass spectrometry, and nuclear magnetic spectroscopy).

First, we randomly split the dataset from ref. 36 into two parts. 5% (38.7k molecules) were used for the case studies and computing the test metrics, and the other 95% were used for training the encoders.

**Representation of spectra for the model.** We represent the spectra as follows: the $^{13}$C NMR as a binary value vector, where there is positional encoding of the chemical shift value (in ppm). The range of the shifts is range 0–300ppm; thus, each bit of the initial vector represents a range of circa 0.59ppm. This choice of chemical shifts covers more than 99% of all molecules in our training dataset.

For $^1$H NMR, each vector has a size of 10,000, where the position in the vector represents the value of the shift (−2–10ppm), and each element represents the intensity of the peak. Before training, all values are normalized between 0 and 1 using min-max scaling. We apply a series of augmentations for the raw spectrum to seem more realistic. These are described in detail in Supplementary Note G.

The IR representation is similar to the $^1$H NMR representation in that the position represents the value of the wavenumber in cm$^{-1}$ and the element's value is the transmittance. We cover the range from 600 cm$^{-1}$ to 3800 cm$^{-1}$ with a vector length of 1600.

The HSQC spectrum is represented as a 512 × 512 matrix, which is reconstructed from the peak information ($^{13}$C centroid, min and max;

[1]H centroid, min and max). To make the HSQC more realistic, we also always include augmentations such as jittering and peak broadening. The code for this procedure is available on our GitHub repository.

## Models

**Joint model architecture.** Our model architecture, inspired by ImageBind[49], allows us to modularly reuse existing models from the literature. To encode SMILES strings, we use the `MolFormer` model from ref. [50], which utilized a transformer architecture[51] with rotary positional encoding[52]. For $^{13}$C NMR we use a multi-layer perceptron (MLP), while for $^1$H NMR and HSQC we opted for a ResNet-style CNN[53] with a self-attention mechanism[51], while for IR a CNN was used. We describe the encoder architectures in more detail in Supplementary Note H (specifically in Supplementary Table 4). Due to varying embedding sizes, we also attach a linear embedding projection layer to each modality encoder. No encoder weights are frozen in contrastive training as this led to a drop in retrieval accuracy in our preliminary experiments. We attribute this to the limited expressivity of a linear layer.

**Training.** The alignment training is based on the InfoNCE[54] contrastive loss defined by the following equation:

$$L_{S,M} = -\log \frac{\exp(\mathbf{q}_i^{\mathsf{T}}\mathbf{k}_i/\tau)}{\exp(\mathbf{q}_i^{\mathsf{T}}\mathbf{k}_i/\tau) + \sum_{i\neq j}\exp(\mathbf{q}_i^{\mathsf{T}}\mathbf{k}_j/\tau)}, \qquad (1)$$

where $S$ represents the spectra embedding and $M$ the molecular representation embedding and $\mathbf{q}$ are the query and $\mathbf{k}$ the key vectors. In practice, we use a symmetric loss ($\mathcal{L}_{symm}$) for training that is defined as follows[49]:

$$\mathcal{L}_{symm} = L_{S,M} + L_{M,S} \qquad (2)$$

**Hyperparameters.** We use a learning rate of $10^{-4}$. For the contrastive loss, we use a temperature ($\tau$) of 0.07 following ref. [55]. The temperature controls the strength of penalties on the negative samples (i.e. spectra and SMILES embeddings that do not match)[56]. We use the early-stopping convergence criterion[57] for the model with a patience of 3 epochs.

**Hardware specification.** The models in this study have been trained using 6 NVIDIA H100-96GB GPUs. For computing the reward used in the genetic algorithm we use one H100-96GB GPU on our local cluster, and one A10G-24GB on the `Modal` API.

## Genetic algorithm

We used an open implementation of the GraphGA algorithm[37]. The aligned models generate different embeddings, $\varepsilon$, of the molecular string representation SMILES, as well as individual embeddings for each spectrum. The reward function $R$, shown in Eq. (3), and used for scoring the candidates consists of two components: an embedding similarity component and a molecular formula component. We define $W$ as the total number of wrong atoms in a new molecule proposed by the GA and $T$ as the correct total number of atoms. By design, the molecular formula penalty is usually lower than the mean of cosine similarities. This allows the GA to operate in an exploratory mode, where constraints are placed on substructure similarities and not a concrete molecule.

$$R = \frac{1}{N}\sum_{x \in \{IR, {}^{13}C, {}^1H, HSQC\}} d_{\cos}(\varepsilon_x, \varepsilon_{SMILES-x}) - \frac{W}{T}, \forall N \in [1,4] \qquad (3)$$

**Pipeline scoring.** We ran three random seeds for fifteen generations for all molecules. After generating the results for individual random seeds, we combine all populations and sort them by their score. The final performance was based on these aggregated results. To understand whether the graph of a generated SMILES matches the original graph, we used a molecular fingerprint with a radius of 6 and a length of 2048[58,59].

**Hardware specifications.** The run of the genetic algorithm driven by the reward function, $R$, takes 2–4 min (on an A10G GPU and faster on a H100 GPU) depending on the size of the molecule. We empirically observed that most runs converge within the first 15 generations of the GA.

## Study cases settings

To generate the spectra for the correct molecules in Fig. 4, we use `MestreNova`, applying the same settings as ref. [38]. They used chloroform-d (CDCl$_3$) as a solvent, with a frequency of 400 MHz. The total sequence length is set at 32,768, which is reduced to 10,000 via linear interpolation.

## Human study

The study was approved by the IRB of the university of Jena. No personal identifiable information (age, gender, etc) were collected because those are not relevant for the comparison of human and model capabilities. Participants were offered the option for participation in a raffle (for the best scoring performance). For this, they could voluntarily self-identify. No compensation was paid to participants.

## Ethics approval

The authors confirm that they have complied with all relevant ethical regulations, according to the Ethics Commission of the Friedrich Schiller University Jena (which decided that the study is ethically safe).

## Reporting summary

Further information on research design is available in the Nature Portfolio Reporting Summary linked to this article.

## Data availability

The checkpoints of the models, the results and the source code can be found as part of the Zenodo record https://doi.org/10.5281/zenodo.14177705[60]. All training, test and benchmarking data (experimental and simulated) can be found in our HuggingFace collection: https://huggingface.co/collections/jablonkagroup/secs-68440633d7438f0d22558879. Source data are provided with this paper.

## Code availability

The source code can be found at https://github.com/lamalab-org/secs and https://doi.org/10.5281/zenodo.14177705. The source code for the app can be found at https://github.com/lamalab-org/secs-app. A test version of the app is available at secs.lamalab.org.

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

## Acknowledgements

The authors thank Erinc Merdivan and Sebastian Starke for fruitful discussions and their Helmholtz AI consultant teams (at HZDR and Helmholtz Munich).

## Author contributions
A.M. and K.M.J. ideated the system. A.M. developed and tested the system and executed all experiments. A.M. also developed the backend code. A.M. and K.M.J. wrote the manuscript. K.M.J. managed the project. L.P. contributed to data analysis and developed custom apps. L.P. also contributed to the design of experimental datasets and the human expert study. All authors revised and approved the final version of the manuscript.

## Funding
This work was funded by the Carl-Zeiss Foundation. In addition, A.M.'s work was partly funded by the Helmholtz Association within the framework of the Helmholtz Foundation Model Initiative (project SOL-AI). Moreover, this work was supported by Helmholtz AI computing resources (HAICORE) of the Helmholtz Association's Initiative and Networking Fund through Helmholtz AI. Parts of this work have been supported by OpenPhilanthropy. K.M.J. is part of the NFDI consortium FAIRmat funded by the Deutsche Forschungsgemeinschaft (DFG, German Research Foundation) - project 460197019. In addition, we thank the participants of our study comparing the performance of SECS against humans and Martiño Ríos García, Meiling Sun, Viktor Weißenborn and Gordan Prastalo for feedback on the manuscript. Open Access funding enabled and organized by Projekt DEAL.

## Competing interests
L.P. is the Chief Scientific Officer of Zakodium Sárl, a company developing scientific software including NMRium, a web-based software for processing NMR spectra. The other authors declare no competing interests.
