## [Transparent Peer Review file · Nature Communications]

End-to-End Multimodal Structure Elucidation from Raw Spectra Combining Contrastive Learning and Evolutionary Algorithms

Corresponding Author: Dr Kevin Maik Jablonka

Version 0:

Reviewer comments:

Reviewer #1

(Remarks to the Author)

In this paper, the authors present spec2struct, a framework designed to predict a list of possible structural candidates from spectroscopic data to automate the process of chemical structure elucidation. The spec2struct is implemented by synergistically combining multimodal embeddings, contrastive learning, and evolutionary algorithms to mimic how expert chemists approach structure determination. In this study, the authors combine the NMR spectra (¹³C-NMR, ¹H-NMR) and IR spectra to improve the prediction accuracy.

Although the chemical structure elucidation may attract the broad audience, the technical advance using different combinations of spectroscopic techniques seems incremental. Also, the paper description is confusing. There are detailed comments that need to be addressed as follows.

1. Figure 2-4 contains too little information. It is advised to merge the information into one figure. And the information in the all figure is not self-evident.
2. More details on the used methodology details and the adopted quality indexes should be given.
3. The basis of the selected threshold AND the relation between the threshold and the prediction probability should be given.
4. In Figure 5, how the R values is obtained, and how much R values can be regarded to be credible. And how the users choose the "correct" structure from the predicted candidate list.
6. The simulated details on the used simulated dataset of ¹³C NMR, ¹H NMR, and IR spectra should be given.
7. Both NMR and IR spectra shown in the example are high-resolution and noiseless. How would the method perform in case of spectral congestion or severe noise? More examples should be illustrated. it would be useful to see more extensive testing on a wider range of compounds to further validate the robustness of the method. Also, is the other NMR techniques applicable to this study, such as pure shift NMR and mD NMR? The method limitation should be further clarified.
8. A more detailed comparison with other state-of-the-art methods in the related field using the general quality indexes should be also given to better position this proposed work.

(Remarks on code availability)

Reviewer #2

(Remarks to the Author)

The authors present a multimodal structure elucidation pipeline that combines latent space alignment with a graph genetic algorithm. The proposed workflow shows good performance in both retrieving molecules from a database and in finding the correct structure starting from a pool of retrieved candidate molecules. This paper should be of interest to a wide range of experimental and computational scientists and thus is highly suitable for the audience of Nature Communications. However, there are some additional analyses that are needed to justify the conclusions in the paper which should be performed before publication:

1. The authors state that they ran the genetic algorithm for at least 10 generations for each molecule. The authors should provide data on the computational speed of the genetic algorithm steps (i.e., the computational hardware that this was run on and the time required on that hardware) and provide some justification for the choice of 10 generations (explicit tabulation of the mean score as a function of the generation number). The authors should also provide a discussion on the scaling of the algorithm's accuracy with respect to the number of generations run for the algorithm. Currently, this detail is only shown as a small sub-plot in Figure 1 so more explicit details are important to include.

2. Based on the results presented for molecular retrieval using the encoder-encoder framework, the chosen candidates for the genetic algorithm tend to be very strong starting points. The authors should test and include a discussion on the performance of the genetic algorithm when the starting population is a worse guess, i.e. if one starts from molecules highly dissimilar to the target by some metric (e.g. Tanimoto similarity), how well does the genetic algorithm perform in finding the correct molecule?

3. Throughout the work, the ¹H NMR input is represented as a vector of 10000 values spanning a shift range of -2 to 10 ppm. The authors should assess the performance of their workflow with varying resolutions of the ¹H NMR in order to better understand how robust their method is against the resolution of the spectrum and the fast varying features typically observed in an ¹H NMR signal.

4. The authors should provide some discussion on how their framework could be adjusted to elucidate components from a mixture of compounds, which is a very common setting in organic synthesis when trying to characterize reaction product mixtures.

(Remarks on code availability)

Reviewer #3

(Remarks to the Author)

(Remarks on code availability)

Version 1:

Reviewer comments:

Reviewer #1

(Remarks to the Author)

I highly appreciate the efforts the author has made in the revision process, and this version has been improved. Although the authors have addressed some concerns pointed out in the previous round, there are still major issues.

1. The limited novelty. Although the automated chemical structure elucidation may attract the broad audience, the technical advance combining two spectroscopic modalities of Nuclear Magnetic Resonance (NMR) and Infrared (IR) spectroscopy is incremental. A mass of automated structure elucidation methods have been proposed separately using NMR spectra, such as:

Ref. 1: The Journal of Physical Chemistry Letters 13, 4924–4933 (2022).

Ref. 2: ChemRxiv: 10.26434/chemrxiv-2023-8wxcz

Ref. 3: ACS Central Science 10, 2162–2170 (2024).

Ref. 4: Anal. Chem., 97, 15736–15742 (2025)

Ref. 5: Chemical Science 11, 4351–4359 (2020).

or using infrared spectroscopy, such as:

Ref. 6: Anal. Chem., 93, 9711–9718 (2021)

Ref. 7: Chemical Science 14, 3600–3609 (2023)

Ref. 8: Digital Discovery 3, 2417–2423 (2024).

Ref.9: Communications Chemistry 7, 1–11 (2024).

Ref. 10: ChemRxiv: 10.26434/chemrxiv-2025-9p2dw. 2025.

Ref. 11: The Journal of Physical Chemistry A 129, 2077–2085 (2025).

Thus, the innovation seems to be manifested only in the combination of two spectroscopic modalities. However, it is not novel, because the multimodal information fusing has been exploring in multiple disciplines, and has been also applied to chemical structure elucidation, such as:

Ref. 12: ACS Omega 10, 12717–12723 (2025).

Ref. 13: Digital Discovery 3, 818–829 (2024).

Other technical issues:

2. The detailed workflows and manuals for Cross-modal retrieval-seeded evolutionary optimization, Physics-informed spectral augmentation pipeline, Calibrated uncertainty quantification, Systematic human-AI performance benchmarking. Also, the Ablation experiments are required.
3. Again, how much R values can be regarded to be credible in automated structure elucidation.
4. In Figure 3B, the prediction accuracy of IR + 13C-NMR + 1H-NMR+HSQC is close to that merely using IR + 13C-NMR. The advantage using combinations of spectroscopic techniques seems to be not obvious.
5. Only six participating chemists in the evaluation to represent human performance.
6. Widespread application validations such as protein analysis, perhaps even more exciting.

Therefore, I would not recommend it in Nature communications.

(Remarks on code availability)

Reviewer #2

(Remarks to the Author)

The authors have satisfactorily addressed all the concerns raised; hence, I would recommend publication.

(Remarks on code availability)

Reviewer #3

(Remarks to the Author)

(Remarks on code availability)

RESPONSE TO THE REVIEWERS

In response to the review of our manuscript, we conducted a very extensive revision in which we also brought in a new coauthor who is a leading expert in NMR spectroscopy and who is leading the development of an open-source NMR processing tool. With this larger team, we:

- developed novel data augmentation schemes to make the system robust on raw experimental data
- performed evaluation on experimental datasets, some of which we compiled just for this purpose
- ran a head-to-head comparison study with experts in NMR spectroscopy on converting raw experimental NMR spectra to structures
- developed multiple front-ends for using our tool as well as for data analysis
- extended the approach to 2D-NMR techniques

We believe that this extensive revision led to a much stronger and widely applicable system that shows multiple unprecedented capabilities.

REVIEWER 1

Reviewer Point P 1.1 — In this paper, the authors present spec2struct, a framework designed to predict a list of possible structural candidates from spectroscopic data to automate the process of chemical structure elucidation. The spec2struct is implemented by synergistically combining multimodal embeddings, contrastive learning, and evolutionary algorithms to mimic how expert chemists approach structure determination. In this study, the authors combine the NMR spectra (^{13}C -NMR, ^1H -NMR) and IR spectra to improve the prediction accuracy. Although the chemical structure elucidation may attract the broad audience, the technical advance using different combinations of spectroscopic techniques seems incremental. Also, the paper description is confusing. There are detailed comments that need to be addressed as follows.

Reply: *We are glad the reviewer finds that the structure elucidation approach “may attract the broad audience”.*

To clarify the advances, we created a new figure that outlines the requirements for a structure elucidation system (Figure R1).

To concretely clarify the technical advances we added a new subsection to the appendix:

To create the first system that fulfills all desired characteristics in Figure R1, we developed several key technical innovations that address fundamental limitations in existing structure elucidation approaches:

- **Multimodal spectroscopic-molecular contrastive alignment:** *We developed the first framework to contrastively align embeddings from heterogeneous spectroscopic modalities (^1H -NMR, ^{13}C -NMR, HSQC, IR) with molecular structure embeddings in a unified latent space. This required designing specialized encoder architectures for each spectroscopic modality (1D CNNs with self-attention for ^1H -NMR, 2D ResNet-style CNNs for HSQC, MLPs for ^{13}C -NMR) while ensuring cross-modal semantic alignment. This technical advance enables direct cross-modal retrieval between any spectroscopic measurement and molecular databases—dramatically expanding searchable chemical space beyond existing spectral databases.*

Figure R1: Requirements for computer-assisted structure-elucidation system and our platform. **A.** We outline five characteristics that would make such a platform most useful. **B.** SECS is an automated platform for getting a ranked list of predictions with database context.

- **Cross-modal retrieval-seeded evolutionary optimization:** We introduced a novel hybrid approach that combines cross-modal retrieval with discrete molecular optimization. Retrieved molecules from our aligned embedding space serve as initial populations for a genetic algorithm operating directly on molecular graphs. This addresses the critical cold-start problem in evolutionary molecular design by providing chemically-relevant starting points rather than random initialization, leading to faster convergence and higher-quality solutions.
- **Physics-informed spectral augmentation pipeline:** We developed a comprehensive augmentation framework that transforms idealized simulated spectra into realistic experimental-like data through physically-motivated perturbations including J-coupling simulation, phase errors, baseline drift, and instrumental noise modeling. This technical innovation enables models trained on simulated data to generalize to real experimental measurements.
- **Calibrated uncertainty quantification:** We established that cosine similarities in our aligned embedding space serve as well-calibrated confidence estimates, with empirical validation showing almost linear correlation between similarity scores and prediction accuracy. This enables reliable uncertainty quantification essential for practical deployment.
- **Systematic human-AI performance benchmarking:** We conducted the first controlled study comparing automated structure elucidation against expert chemists on identical molecular identification tasks, establishing quantitative baselines for human-level performance in this domain.

These technical advances synergistically address the five key limitations of existing approaches: requirement for manual preprocessing, inability to combine multiple spectroscopic modalities, lack of confidence estimates, restriction to database compounds, and absence of relevant contextual information for interpretation.

Figure R2: New figure with retrieval results (Figure 3 in revised manuscript).

Reviewer Point P 1.2 — 1. Figure 2-4 contains too little information. It is advised to merge the information into one figure. And the information in the all figure is not self-evident. We also provided more detailed captions for each figure.

Reply: *We merged the figures into a new retrieval results figure (Figure R2).*

Reviewer Point P 1.3 — 2. More details on the used methodology details and the adopted quality indexes should be given.

Reply: *We created a new figure that explains the methodology in more details (Figure R3).*

Figure R3: Overview of the SECS workflow. **A.** The first step is the training of contrastively aligned encoders for various spectra and the SMILES. **B.** Toward this goal, contrastively trained models between spectra and molecules are used to query compounds that have similar compositions. This enriches the candidate pool for the starting population. SECS then embeds all the retrieved molecules and ranks the compounds based on a multimodal similarity score between the molecule embeddings and the spectra embeddings. **C.** We then run an evolutionary algorithm (GraphGA) on the top-ranked molecules in step **B**. The output of the ranked list might contain a mixture of novel and retrieved compounds. All compounds and spectra are shown for illustration purposes.

In addition, we expanded both the methods section and appendix to include more methodological detail, such as entire section on data augmentation, a section on retrieval metrics, and detailed overviews over the model architectures.

Reviewer Point P 1.4 — 3. The basis of the selected threshold AND the relation between the threshold and the prediction probability should be given.

Reply: See response to Reviewer Point P 1.5.

Reviewer Point P 1.5 — 4. In Figure 5, how the R values is obtained, and how much R values can be regarded to be credible. And how the users choose the “correct” structure from the predicted candidate list.

Reply: We now clarify in the caption of Figure 3C (of the revised version of the manuscript, previous figure 5) the meaning of R with reference to the equation defining it

The correlation between the score of the pipeline R , demonstrated in eq. (3) against the average performance of the pipeline. We observe a well-calibrated system.

In addition, we give an example of how to interpret the score in the main text

Figure 3C shows that if, for example, the score is above a min-max scaled threshold of 0.94, there is a probability of around 94 % that the prediction made by the genetic algorithm is correct for examples that are drawn from the same distribution as the ones of the evaluation data.

Regarding the choice of the correct structure we now added a discussion in the main text:

In an autonomous setting, one could thus automatically select the top-ranked candidate. However, experts can often benefit from the additional context provided by returning more than one candidate, wherefore we typically return 20 candidates in the SECS application.

Reviewer Point P 1.6 — 6. The simulated details on the used simulated dataset of ^{13}C NMR, ^1H NMR, and IR spectra should be given.

Reply: In the methods/appendix section we now provide a table listing all datasets we used in the study and the sources (Table R1):

Table R1: An overview of all the used datasets in this study.

Dataset	Modalities	Count	Source
Simulated dataset (training)	^1H NMR, ^{13}C NMR, IR, HSQC	705k	Alberts et al. ¹
Simulated dataset (test)	^1H NMR, ^{13}C NMR, IR, HSQC	38.8k	Alberts et al. ¹
Multimodal experimental dataset for finetuning	^1H NMR, ^{13}C NMR	2,37 ^1H NMR, 19,654 ^{13}C NMR	in-house + NMRshiftDB
Multimodal experimental dataset for testing	^1H NMR, ^{13}C NMR	34 each	in-house + NMRshiftDB
Chemotion dataset	^1H NMR	1.49k	mined

We reference details in a new subsection "Study cases settings" (in the Methods section):

To generate the spectra for the correct molecules in Figure 4, we use MestreNova, applying the same settings as Alberts et al.² They used deuterated chloroform as a solvent, with a frequency of 400 MHz. The total sequence length is set at 32,768 and reduced to 10,000 via linear interpolation.

Reviewer Point P 1.7 — 7. Both NMR and IR spectra shown in the example are high-resolution and noiseless. How would the method perform in case of spectral congestion or severe noise?

Reply: In the revision, we approached this in two ways. First, we introduced new data techniques and dedicated an entire section in the appendix to it. In addition, to studying the effect of data augmentation, we also studied the effect of resolution changes. We summarize the main trends in a new figure (Figure R4):

(a) Impact of different augmentation levels on experimental data retrieval performance (within a set of 2.37k spectra-molecule pairs). The best performance is achieved when custom augmentation levels are chosen.

(b) Impact of different spectra resolutions on the recall@1 among simulated spectra. The models are trained for five epochs and tested on the 38.8k test set of simulated spectra. The resolution has no effect on the overall recall.

Figure R4: Ablating the effects of augmentation on the experimental data, and the effect of the ^1H NMR resolution on the simulated test set retrieval.

In addition, we also performed evaluation on experimental datasets and dedicate a new section in the main text to it

SECS on experimental data

To further assess the performance of SECS, we tested its performance in multiple experimental case studies.

In-house dataset

As a first case study, we compiled an in-house dataset of 34 molecules with experimental ^1H NMR and ^{13}C NMR, for which the molecules are not in our training set.

Figure R5 shows the performance of different versions of our pipeline. In the top row, we show the performance on simulated spectra and in the bottom row on experimental spectra. While we find decent retrieval performance with our model trained on the simulation data from Alberts et al.,¹ we do not find any meaningful results using the experimental inputs.

However, applying data augmentation as described in Figure R5 leads to a leap in performance on both the simulated and experimental datasets.

By further finetuning the model on a disjoint training set of 2.37k molecules along with experimental spectra generated in-house, we could further improve the performance on experimental data.

We observe that the impact of the augmentation is major, with a performance jump on experimental data from 0% (top-1) for purely simulated spectra to 38.2% for the model trained on augmented spectra, to finally 55.8% at top-1 for the model finetuned on experimental data. We also show that by adding ^{13}C NMR (from NMRshiftDB^{3,4}) the performance can increase significantly to 88.2% top-1 and 97.1% top-3 performance.

Recently published NMR spectra

As a challenging real-world case study, we retrieved spectra that have recently been published in the Chemotion repository.^{5,6} None of the spectra appeared in our training sets and, as Figure I.11 shows, they expand the chemical space we test (see also molecular weight distribution in Figure I.12). We find that SECS also shows a high performance on this dataset, correctly predicting 27.9% for all compounds matching the formula $\text{C}_{3-15}\text{H}_{4-42}\text{O}_{0-5}\text{N}_{0-5}\text{Cl}_{0-5}$ containing 15 or less heavy atoms and returning the correct compound within the first 20 predictions in 58.5% of the cases (323 compounds). When we consider the entire dataset (1486 compounds) the top-1 performance is 10.3% and the top-20 performance is 24.8%.

Figure R5: The performance of SECS on ^1H NMR, ^{13}C NMR and a combination of the two spectral methodologies. Here we present the result for 34 molecules for which we had all three spectra: simulated ^1H NMR, experimental ^1H NMR and experimental ^{13}C NMR. The title of each subplot indicates on each type of data each model has been trained, with the legend indicating for what spectra types the results are shown. For example, simulation-pretrained and green indicate that the underlying model has been pretrained on simulated spectra but tested on experimental spectra. “Finetuned” indicates that the augmentation-based model has been trained on an additional experimental dataset.

Comparison with human performance

To contextualize SECS performance against human expertise, we conducted a controlled study comparing our tool with experienced chemists on molecular structure elucidation tasks (see Section 2.1 for details). We selected 20 challenging spectra from the Chemotion dataset at random (the molecules should follow the molecular formula limits stipulated above) and developed a custom web application where participants analyzed ^1H NMR spectra. We tasked participants with drawing the chemical structures of unknown compounds using molecular editing tools. For each compound, we provided the correct molecular formula and double-bond equivalents and provided immediate feedback on each proposed structure regarding its consistency with these parameters. All participants could use additional resources of their choosing, including databases such as NMRshiftDB. The challenge proved formidable even for experts: the top-performing participant—a scientist specializing in NMR spectroscopy for over two decades—solved fewer than half of the problems, with participants averaging 13.2 h to complete all 20 tasks.

Figure R6 demonstrates that SECS achieves expert-level performance on these challenging molecular identification tasks. In head-to-head comparisons, SECS outperformed 67% of participating chemists (4 of 6 participants), ranking at the 64th percentile overall with a $\Delta = +0.8$ advantage over the human average. These results establish that SECS not only automates structure elucidation but does so at a level competitive with domain experts.

Figure R6: SECS demonstrates expert-level performance in molecular structure elucidation from NMR spectra. **A.** Individual performance scores for human chemists (circles, $n = 6$) and SECS (brown diamond) on 20 molecular formula identification challenges. Horizontal lines indicate group means. SECS outperforms the human average by $\Delta = +0.8$ correct answers. **B.** Cumulative performance distribution. SECS achieves 64th percentile performance among all participants. Head-to-head comparison reveals the AI model outperforms 4 of 6 human chemists (67%). The three top-scoring humans are NMR experts with experience ranging from multiple years to multiple decades.

Reviewer Point P 1.8 — More examples should be illustrated. it would be useful to see more extensive testing on a wider range of compounds to further validate the robustness of the method.

Reply: See response to Reviewer Point P 1.7.

Reviewer Point P 1.9 — Also, is the other NMR techniques applicable to this study, such as pure shift NMR and mD NMR? The method limitation should be further clarified.

Reply: In the revised version, we also demonstrate how 2D NMR (specifically HSQC) can be incorporated to improve the predictions of the model.

Reviewer Point P 1.10 — 8. A more detailed comparison with other state-of-the-art methods in the related field using the general quality indexes should be also given to better position this proposed work.

Reply: We believe that the combination of features our tool provides is unique (see also response to Reviewer Point P 1.1). In the Introduction of the revised version, we make the following point regarding past literature approaches:

these techniques still ignore the reality that chemists must flexibly combine many different analytical techniques for structure elucidation and operate on raw instrument outputs

However, we still perform some benchmarking to what we believe is the closest competing approach in Figure 3B of the revised manuscript, where the caption reads:

Structure elucidation performance as a function of the number of spectra used against the literature baseline from Alberts et al.²(top-1).

To encourage further systematic comparisons, we openly release all benchmark datasets we have created for this work on HuggingFace, as described in the data availability section:

The code and the simulation data can be found at <https://github.com/lamalab-org/secs>. The checkpoints of the models, and the results can be found can be found as part of the Zenodo record [10.5281/zenodo.14177705](https://zenodo.org/record/105281). A test version of the app is available at secs.lamalab.org.

All training, test and benchmarking data (experimental and simulated) can be found in our HuggingFace collection: <https://huggingface.co/collections/jablunkagroup/secs-68440633d7438f0d22558879>.

In addition, we also performed a novel study in which we compared the performance of our system with the one of human experts.

REVIEWER 2

Reviewer Point P 2.1 — The authors present a multimodal structure elucidation pipeline that combines latent space alignment with a graph genetic algorithm. The proposed workflow shows good performance in both retrieving molecules from a database and in finding the correct structure starting from a pool of retrieved candidate molecules. This paper should be of interest to a wide range of experimental and computational scientists and thus is highly suitable for the audience of Nature Communications. However, there are some additional analyses that are needed to justify the conclusions in the paper which should be performed before publication:

Reply: *We are grateful to read that the reviewer recognized that this paper is of interest to a wide range of scientists.*

Reviewer Point P 2.2 — 1. The authors state that they ran the genetic algorithm for at least 10 generations for each molecule. The authors should provide data on the computational speed of the genetic algorithm steps (i.e., the computational hardware that this was run on and the time required on that hardware) and provide some justification for the choice of 10 generations (explicit tabulation of the mean score as a function of the generation number). The authors should also provide a discussion on the scaling of the algorithm's accuracy with respect to the number of generations run for the algorithm. Currently, this detail is only shown as a small sub-plot in Figure 1 so more explicit details are important to include.

Reply: *In Figure 3D of the revised version of the main text (see Figure R2), we now demonstrate the relationship between the number of seeds and the performance of the pipeline.*

The timing numbers and the convergence are described in Methods (Section 4.3 under the paragraph "Hardware specifications"):

The run of the genetic algorithm driven by the reward function, R, takes 2–4 minutes (on an A10G GPU and faster on a H100 GPU) depending on the size of the molecule. We empirically observed that most runs converge within the first 15 generations of the GA.

Alongside empirical observations and hardware specifications we now include a table that shows the mean and max score as a function of the generation, also including the standard deviations for both metrics.

In Table R2, we present the relationship between the number of generations and the max, mean scores and the respective standard deviations. We do observe that the GA most frequently converges within the first few generations.

Reviewer Point P 2.3 — 2. Based on the results presented for molecular retrieval using the encoder-encoder framework, the chosen candidates for the genetic algorithm tend to be very strong starting points. The authors should test and include a discussion on the performance of the genetic algorithm when the starting population is a worse guess, i.e. if one starts from molecules highly dissimilar to the target by some metric (e.g. Tanimoto similarity), how well does the genetic algorithm perform in finding the correct molecule?

Reply: *In Appendix H we study this:*

We analyzed several parameters, such as the maximum Tanimoto similarity of any molecule from the initial population to the target molecule, the maximum

Table R2: Tabulated scores per generation. The max and mean score across generations 1 to 15 along with their respective standard deviation.

Generation	Mean score	Max score
1	0.027 ± 0.098	0.777 ± 0.122
2	0.027 ± 0.092	0.813 ± 0.112
3	0.030 ± 0.089	0.833 ± 0.102
4	0.034 ± 0.088	0.846 ± 0.093
5	0.039 ± 0.088	0.854 ± 0.086
6	0.043 ± 0.088	0.861 ± 0.081
7	0.065 ± 0.093	0.865 ± 0.077
8	0.098 ± 0.097	0.868 ± 0.075
9	0.125 ± 0.099	0.871 ± 0.071
10	0.149 ± 0.101	0.874 ± 0.068
11	0.172 ± 0.103	0.876 ± 0.065
12	0.194 ± 0.105	0.878 ± 0.064
13	0.215 ± 0.107	0.879 ± 0.062
14	0.234 ± 0.109	0.880 ± 0.061
15	0.253 ± 0.111	0.881 ± 0.060

cosine similarity between the spectra embedding and any compound (for the purpose of the experiments the correct molecule is removed) in the initial population to the correct molecule, and the Bertz complexity.^{7,8} The results of these analyses are shown in Figure R7. The performance of our algorithm directly depends on all of these factors. For instance, the distribution of Bertz complexities is skewed towards higher values when the correct molecule is not found. The maximum Tanimoto similarity of the compounds in the initial population also plays a pivotal role in determining whether the right compound is found. The maximum initial Tanimoto similarities are skewed towards the range 0.8 – 1.

Figure R7: The distribution of correct vs incorrect predictions (recall@1=True or recall@1=False). The metrics used to understand the domain of applicability are (a) Tanimoto similarity, (b) cosine similarity, and (c) Bertz CT of the correct molecule. The dataset used for this analysis is the same test set used to create Figure 2 in the main text with the performance for three different seeds and four spectra (IR, ¹H NMR, ¹³C NMR, HSQC).

Reviewer Point P 2.4 — 3. Throughout the work, the ¹H NMR input is represented as a vector of 10000 values spanning a shift range of -2 to 10 ppm. The authors should assess the performance of their workflow with varying resolutions of the ¹H NMR in order to better understand how robust their method is against the resolution of the spectrum and the fast varying features typically observed in an ¹H NMR signal.

Reply: See response to reviewer point P 1.7.

Reviewer Point P 2.5 — 4. The authors should provide some discussion on how their framework could be adjusted to elucidate components from a mixture of compounds, which is a very common setting in organic synthesis when trying to characterize reaction product mixtures.

Reply: *We extended the discussion about future work in the main text to read*

SECS currently is also not optimized to handle mixtures. However, this might be implemented in future work with, for example, non-negative matrix factorization (NMF)⁹ or iterative application of the algorithm.¹⁰

REFERENCES

- [1] Alberts, M.; Schilter, O.; Zipoli, F.; Hartrampf, N.; Laino, T. *arXiv preprint arXiv: 2407.17492* **2024**,
- [2] Alberts, M.; Schilter, O.; Hartrampf, N.; Laino, T. A Multimodal Transformer Model for comprehensive Structure Elucidation. American Chemical Society (ACS) Fall Meeting. 2024.
- [3] Steinbeck, C.; Kuhn, S. *Phytochemistry* **2004**, *65*, 2711–2717.
- [4] Kuhn, S.; Kolshorn, H.; Steinbeck, C.; Schlörer, N. *Magnetic Resonance in Chemistry* **2024**, *62*, 74–83.
- [5] Tremouilhac, P.; Lin, C.-L.; Huang, P.-C.; Huang, Y.-C.; Nguyen, A.; Jung, N.; Bach, F.; Ulrich, R.; Neumair, B.; Streit, A.; others *Angewandte Chemie (International ed. in English)* **2020**, *59*, 22771.
- [6] Tremouilhac, P.; Huang, P.-C.; Lin, C.-L.; Huang, Y.-C.; Nguyen, A.; Jung, N.; Bach, F.; Bräse, S. *Chemistry-Methods* **2021**, *1*, 8–11.
- [7] Tanimoto, T. T. **1958**,
- [8] Bertz, S. H. *Journal of the American Chemical Society* **1981**, *103*, 3599–3601.
- [9] Snyder, D. A.; Zhang, F.; Robinette, S. L.; Bruschweiler-Li, L.; Bruschweiler, R. *The Journal of Chemical Physics* **2008**, *128*, 052313.
- [10] Szymanski, N. J.; Bartel, C. J.; Zeng, Y.; Tu, Q.; Ceder, G. *Chemistry of Materials* **2021**, *33*, 4204–4215.

RESPONSE TO THE REVIEWERS

We are grateful for the opportunity to submit a further revision of our manuscript. We appreciate that Reviewers 2 and 3 now fully recommend publication. We address below the remaining points raised by Reviewer 1.

Overall, we performed the following changes to the manuscript: We

- *Strengthened the contextualization with current literature.* We expanded the Discussion to compare SECS explicitly with recent structure elucidation approaches, including those suggested by Reviewer 1. We now clearly delineate how SECS differs conceptually from:
 - unimodal models that decode a single spectrum into a structure
 - multimodal models that fuse several spectra within a single neural network but do not provide cross-modal retrieval, calibrated uncertainty, or adaptable structure generation.

We also highlight the fact that most, if not all, current CASE solutions require manual preprocessing of spectra and thus still pose bottlenecks in automated workflows. To emphasize this, we also updated the title of the manuscript to “End-to-End Multimodal Structure Elucidation from Raw Spectra Rivaling Expert Performance”.

- *Clarified the significance of combining multiple spectroscopic modalities.* We added additional context to the Introduction and added additional analyses to the Supplementary Material. We also emphasize that SECS is designed to flexibly handle any subset of modalities available in practice (IR, ^1H NMR, ^{13}C NMR, HSQC), rather than requiring a fixed combination.
- *Provided clearer workflows and ablation studies.* We now added a workflow diagram to the appendix and more clearly call out ablation studies. We also added a new section defining metrics.
- *Clarified the interpretation of the reward R and confidence.* In the section discussing Fig. 3C we now explicitly state the empirical mapping between the min-max scaled reward and accuracy, and give practical guidance on how to interpret it as a practitioner.
- *Tempered and contextualized the human-performance claims.* We changed the title of the manuscript to reflect that the performance comparison was made based on a limited subset of experts to also added relevant context in the abstract and discussion sections.

We believe these changes address the remaining concerns while preserving the core contributions of SECS:

1. novel “retrieve, then refine” ansatz for structure elucidation
2. end-to-end elucidation capabilities based on raw experimental data
3. contrastive multimodal alignment between raw spectra and molecular structures
4. cross-modal retrieval-seeded evolutionary optimization that can propose novel structures
5. physics-informed sim-to-experiment transfer (via data augmentation and fine-tuning)
6. calibrated confidence estimates
7. a controlled comparison to expert chemists

8. reusable code and a web app

We want to reiterate that the end-to-end performance on raw experimental spectra is difficult to achieve. As Devata et al.¹ highlight, the performance of existing pipelines is often “heavily dependent on the correctness of the peak-picking step”. Our approach does not require any such manual processing. In the last revision, we introduced novel augmentation procedures to enable predictions on experimental data, for which we also curated multiple novel datasets to, for the first time, enable thorough assessment on raw experimental data. We also report the generalization performance on OOD data. To allow hands-on testing, we also provide a web app and reusable code.

REVIEWER 1

I highly appreciate the efforts the author has made in the revision process, and this version has been improved. Although the authors have addressed some concerns pointed out in the previous round, there are still major issues.

Reviewer Point P 1.1 — The limited novelty. Although the automated chemical structure elucidation may attract the broad audience, the technical advance combining two spectroscopic modalities of Nuclear Magnetic Resonance (NMR) and Infrared (IR) spectroscopy is incremental. A mass of automated structure elucidation methods have been proposed separately using NMR spectra, such as:

Ref. 1: *The Journal of Physical Chemistry Letters* 13, 4924–4933 (2022).

Ref. 2: ChemRxiv: 10.26434/chemrxiv-2023-8wxcz

Ref. 3: *ACS Central Science* 10, 2162–2170 (2024).

Ref. 4: *Anal. Chem.*, 97, 15736–15742 (2025)

Ref. 5: *Chemical Science* 11, 4351–4359 (2020).

or using infrared spectroscopy, such as:

Ref. 6: *Anal. Chem.*, 93, 9711–9718 (2021)

Ref. 7: *Chemical Science* 14, 3600–3609 (2023)

Ref. 8: *Digital Discovery* 3, 2417–2423 (2024).

Ref. 9: *Communications Chemistry* 7, 1–11 (2024).

Ref. 10: ChemRxiv: 10.26434/chemrxiv-2025-9p2dw. 2025.

Ref. 11: *The Journal of Physical Chemistry A* 129, 2077–2085 (2025).

Thus, the innovation seems to be manifested only in the combination of two spectroscopic modalities. However, it is not novel, because the multimodal information fusing has been exploring in multiple disciplines, and has been also applied to chemical structure elucidation, such as:

Ref. 12: *ACS Omega* 10, 12717–12723 (2025).

Ref. 13: *Digital Discovery* 3, 818–829 (2024).

Reply: *We appreciate the reviewer’s remark that automated chemical structure elucidation from spectra is likely to attract a broad audience, and we fully agree. Our central claim is not that any single component of SECS is, in isolation, a completely new algorithmic primitive, but that this application only becomes practically viable when several technical innovations are combined in one coherent system.*

In the introduction, we added several clarifying sentences related to all the innovations in this work:

In this work, we report a system (SECS - structure elucidation from chemical spectra) that produces a ranked list of structural candidates based on flexible combinations of raw experimental inputs. It can adapt to new chemical domains and provides relevant context that aids interpretation. It is based on a novel “retrieve, then refine” ansatz that provides a chemically informed, and customizable prior for a generative approach. As we show in more detail in Table R1, SECS is, to our knowledge, the only available CASE system that does so.

To make the novelty aspect even more explicit, we have added a new comparison table (Table S1) in the Appendix that includes all works cited by the reviewer, as well as closely related recent methods:

We also compare SECS to past attempts to build CASE systems. In Table R1, we delineate the differences, exposing some fundamental limitations of past approaches. In this table, 9 different criteria are evaluated, that can be placed in three distinct groups: overall system capabilities (dereplication, elucidation,

end-to-end, calibrated uncertainty, adaptable chemical space), limitations (max heavy atom count, easy to add modalities) and evaluation (expert comparison, raw experiment evaluation). Our system stands out as the most robust approach for structure elucidation, encompassing all the attributes of a complete CASE system. On the evaluation side, this work is the only one that compares to NMR experts, providing us with an approximate upper bound for the human performance on a narrow dataset (9/20 molecules solved by the most experienced NMR expert with circa 20 years of experience).

Below we define the terms from Table R1 one-by-one:

- **Dereplication** The ability of a CASE system to identify a molecule based on already collected samples.
- **Elucidation** The ability of a CASE system to propose new structures.
- **End-to-end** The ability of a CASE system to start directly from the raw data and end with proposing the final candidates without manual inputs / interventions such as lists of picked peaks.
- **Calibrated uncertainty** The ability of the system to provide confidence estimates for its own predictions.
- **Adaptable chemical space** The ability of the system to adapt its proposals depending on an input database.
- **Max heavy atoms** This represents a limitation denoting the maximum number of heavy atoms in a target molecule.
- **Easy to add modalities** The ability of the system to easily incorporate new types of spectra without retraining any of the existing models.
- **Expert comparison** The existence of a comparison to expert spectroscopists.
- **Raw experiment evaluation** The system should be evaluated on raw (realistic) experimental data, and not idealized spectroscopic analysis with only a solvent and the target molecule. Such raw data is more common in the chemistry lab, where chemical reactions or natural product mixtures are routinely analyzed.

Table R1: Comparing the technical capabilities of SECS with prior works on structure elucidation. We define the fundamental properties of a complete CASE system and compare SECS to other approaches using these properties as criteria. HA = Heavy Atoms.

System	Overall System Capabilities					Limitations		Evaluation	
	Dereplication	Elucidation	End-to-end	Calibrated uncertainty	Adaptable chem. space	Max HA count in development	Easy to add modalities	Expert comparison	Raw experiments evaluation
SECS (This work)	✓	✓	✓	✓	✓	35	✓	✓	✓
Structure confirmation									
DP4-AI ²	×	×	×	×	×	N/A	×	×	✓
Partial structure identification									
Enders et al. ³	×	×	×	×	×	N/A	×	×	✓
Lee et al. ⁴	×	×	×	×	×	N/A	×	×	×
Alberts et al. (2025) ⁵	×	×	×	×	×	35	×	×	×
Unimodal elucidation									
Sridharan et al. ⁶	×	✓	×	×	×	10	×	×	×
Alberts et al. (2025) ⁷	×	✓	×	×	×	35	×	×	✓
Alberts et al. (2024) ⁸	×	✓	×	×	×	35	×	×	✓
Kanakala et al. ⁹	✓	✓	×	×	×	9	×	×	×
Wu et al. ¹⁰	×	✓	×	×	×	13	×	×	×
Multimodal elucidation									
Alberts et al. (2023) ¹¹	×	✓	×	×	×	35	×	×	×
Pesek et al. ¹²	×	✓	×	×	×	19	×	×	×
DeepSPIN ¹	×	✓	✓	×	×	10	×	×	×
Sherlock ¹³	✓	✓	×	×	×	40	×	×	×
Hu et al. ¹⁴	×	✓	×	×	×	19	×	×	×
MultiModalTransformer ¹⁵	×	✓	×	×	×	35 (500 Da)	×	×	×

Moreover, we now point out the flexibility of our approach, which we believe to be essential for real-world applications but is not enabled by prior approaches.

Existing multimodal approaches, however, fail to recognize that a system must flexibly combine modalities, as not all of them might be needed or available. For instance, they might require all spectra a model has been trained on as input during inference, inevitably wasting resources for simple systems. They are also not adaptable to easily (without retraining the full system) incorporate new techniques that might be needed to fully resolve a new set of spectroscopic problems.

Reviewer Point P 1.2 — Other technical issues:

The detailed workflows and manuals for Cross-modal retrieval-seeded evolutionary optimization, Physics-informed spectral augmentation pipeline, Calibrated uncertainty quantification, Systematic human-AI performance benchmarking. Also, the Ablation experiments are required.

Reply: To further facilitate the dissemination of the techniques used to build SECS, we provide additional pointers, and add new explanations for the workflow & human study, but also additional analyses for the results.

To explain the cross-modal retrieval-seeded evolutionary optimization, we add a detailed diagram in Appendix C.

SECS utilizes a novel “retrieve, then refine” approach to flexibly incorporate inductive biases (via the reference database) but also allow for scalability. Figure R1 gives an overview of the SECS workflow.

Figure R1: Overview of the SECS structure elucidation pipeline. SECS processes raw experimental spectra without manual preprocessing. **(1) Input:** Any combination of ¹H NMR, ¹³C NMR, HSQC, and IR spectra; missing modalities are handled gracefully. **(2) Encoding:** Modality-specific neural networks (architectures in ??) map raw spectra to embedding vectors \mathbf{z}_m . **(3) Cross-modal retrieval:** Per-modality cosine similarities between spectral embeddings and precomputed molecular embeddings $\mathbf{z}_{\text{mol}}^{(i)}$ are summed to rank database candidates, returning the top-k matches. **(4) Optimization:** Retrieved candidates seed a genetic algorithm (GraphGA) that refines structures using summed embedding similarity as the fitness function R . **(5) Output:** Ranked structure predictions with calibrated confidence scores. The dashed feedback loop indicates iterative re-embedding of proposed structures during optimization. This architecture enables database adaptation without retraining—users can substitute the reference database to target different chemical spaces.

Moreover, we also have a graphic explanation for the augmentation pipeline. For an updated explanation of the calibrated uncertainty (via the reward R) see Point 1.3. Additionally, the description of the human study and snapshots of the web app can be found in expanded appendices.

We reorganized all of the ablations into one single section (Appendix K). Here, we additionally ablate the benefit of adding modalities (see Point 1.4 for details), on top of ablations

from the previous version such as: the effect of starting from a single vs. multiple molecular formula (Appendix K.1) and, resolution and augmentation levels (Appendix K.3).

Additionally, Figure 5 in the main text ablates the presence or absence of augmentation in the context of raw experiment performance.

Reviewer Point P 1.3 — Again, how much R values can be regarded to be credible in automated structure elucidation.

Reply: We try to provide, for additional clarity, an updated explanation that refers to R as the reward:

for example, the reward, R, is above a min-max scaled threshold of 0.94, there is a probability of around 94 % that the prediction made by the genetic algorithm is correct for examples that are drawn from the same distribution as the ones of the evaluation data

Additionally, we clarify all metrics in a newly created section (Appendix E):

Recall@K

$$\text{Recall@K} = \frac{1}{N} \sum_{i=1}^N \mathbf{1}[\text{rank}_i \leq K] \quad (1)$$

where N is the total number of queries, rank_i is the rank of the relevant item for query i, and $\mathbf{1}[\cdot]$ is the indicator function. Essentially, it's the fraction of queries where the correct answer appears in the top K results

Mean Reciprocal Rank (MRR@K)

$$\text{MRR@K} = \frac{1}{N} \sum_{i=1}^N \begin{cases} \frac{1}{\text{rank}_i} & \text{if } \text{rank}_i \leq K \\ 0 & \text{otherwise} \end{cases} \quad (2)$$

Special Case: K = 1

At K = 1, both metrics reduce to accuracy:

$$\text{Recall@1} = \text{MRR@1} = \frac{1}{N} \sum_{i=1}^N \mathbf{1}[\text{rank}_i = 1] \quad (3)$$

Fraction Elucidated (Top-K) The fraction of molecules for which the correct structure is identified within the top K predictions of the full pipeline (retrieval + genetic algorithm):

$$\text{Fraction Elucidated@K} = \frac{1}{N} \sum_{i=1}^N \mathbf{1}[\text{rank}_i \leq K] \quad (4)$$

where N is the total number of test molecules and rank_i is the rank of the correct structure in the final sorted candidate list for molecule i. This metric differs from retrieval recall in that it evaluates the complete SECS pipeline output, which includes both retrieved database compounds and novel structures proposed by the genetic algorithm.

Multimodal Reward Function (R) The reward function used to score candidates in the genetic algorithm:

$$R = \frac{1}{M} \sum_{x \in S} d_{\cos}(\epsilon_x, \epsilon_{\text{SMILES}}) - \frac{W}{T} \quad (5)$$

where $S \subseteq \{\text{IR}, {}^{13}\text{C}, {}^1\text{H}, \text{HSQC}\}$ is the set of available spectroscopic modalities, $M = |S|$ is the number of modalities used, d_{\cos} is cosine similarity, ϵ_x and

ϵ_{SMILES} are the embeddings for spectrum x and the molecular SMILES respectively, W is the number of wrong atoms in the proposed molecule, and T is the total number of atoms in the correct molecular formula.

R can also be used as a measure of how calibrated the system is with respect to the total fraction of elucidated structures (**calibrated uncertainty**), which we also show in Figure 3C. To assess calibration, predictions are grouped into quantile bins based on their reward scores R . For each bin b , the empirical accuracy is computed as:

$$\text{Accuracy}(b) = \frac{1}{|b|} \sum_{i \in b} \mathbf{1}[\text{rank}_i = 1] \quad (6)$$

where $|b|$ is the number of samples in bin b . A well-calibrated system exhibits a linear relationship between the mean reward score of each bin and its corresponding fraction of correctly elucidated spectra, such that higher confidence scores reliably indicate higher accuracy.

Reviewer Point P 1.4 — In Figure 3B, the prediction accuracy of IR + ^{13}C -NMR + ^1H -NMR+HSQC is close to that merely using IR + ^{13}C -NMR. The advantage using combinations of spectroscopic techniques seems to be not obvious.

Reply: We provide additional evidence for the benefit of adding new modalities in Appendix K, besides looking at top-1 and top-5 only. In the new plot (Figure R2) we show the average rank of prediction when the correct molecule is found in the final list of proposals. We see that adding new modalities reduces the average rank.

In this section, we provide additional evidence for the benefit of adding new modalities, besides the simpler metric of ranking within top-N candidates. From Figure 3, we see a boost in performance with every added modality, but the exact rank of each correct prediction is also an important aspect.

In addition, we contextualize it in the main text

That is, the combination of modalities allows us to solve problems that none of the modalities could solve alone (see reduction of extreme ranks in Figure K.11 with increasing number of modalities).

Reviewer Point P 1.5 — Only six participating chemists in the evaluation to represent human performance.

Reply: This concern is valid.

Nevertheless, we believe that 120 human comparison data points provide grounds to conclude that SECS outperforms PhD students and approaches expert-level performance on this benchmark. The inclusion of experts with decades of experience provides a strong upper boundary, but limits also the scale of the study since experts took on average 13.2h to complete the assignment.

Given that this evaluation is the first of its kind, we also collect additional feedback via a dedicated feedback form on the web app.

We also included a TOST statistical test (Two One-Sided Test) to show that the human performance and the system performance are not different from each other within a margin of $\pm 20\%$:

(on an equivalence Two One-Sided Tests with a margin of 20% we obtain $p_{\text{TOST}} = 0.0287$ indicating that within this margin the human performance and SECS performance do not significantly differ)

Additionally, we modify how we refer to the human benchmark in the abstract and reframe it as a pilot study:

Figure R2: The effect of incrementally adding new modalities. The rank subplot follows the same order of addition as in Figure 3B in the main text). We observe that the addition of new modalities limits the amount of extreme ranks of the correct prediction, always being below 20 for all four modalities (when the correct molecule is found) indicating that our system becomes increasingly confident.

On challenging molecular identification tasks, SECS matches expert chemist performance in head-to-head comparisons in a pilot study.

In the results section, we also make this clarification more prominent:

While, due to the time required for experts to solve the tasks, our sample size is limited, these results indicate that SECS not only automates structure elucidation but does so at a level competitive with the domain experts in our study. To enable further validation by the community, we openly release our system and benchmark datasets.

Reviewer Point P 1.6 — Widespread application validations such as protein analysis, perhaps even more exciting.

Therefore, I would not recommend it in Nature communications.

Reply: *We now added this to the limitations section (part of the results) in order to clarify important future directions for our system:*

Systems like SECS might also be used for advanced problems like protein structure elucidation, which involves different spectroscopic signatures (e.g., NOESY), databases (e.g., PDB), and validation approaches (e.g., Ramachandran analysis). Future work is also necessary to elucidate mixtures, and might be implemented in future work with, for example, non-negative matrix factorization (NMF)¹⁶ or iterative application of the algorithm.¹⁷

REFERENCES

- [1] Devata, S.; Sridharan, B.; Mehta, S.; Pathak, Y.; Laghuvarapu, S.; Varma, G.; Priyakumar, U. D. *Digital Discovery* **2024**, Publisher: RSC.
- [2] Howarth, A.; Ermanis, K.; Goodman, J. M. *Chemical science* **2020**, *11*, 4351–4359.
- [3] Enders, A. A.; North, N. M.; Fensore, C. M.; Velez-Alvarez, J.; Allen, H. C. *Analytical Chemistry* **2021**, *93*, 9711–9718.
- [4] Lee, G.; Shim, H.; Cho, J.; Choi, S.-I. *ACS omega* **2025**, *10*, 12717–12723.
- [5] Alberts, M.; Hartrampf, N.; Laino, T. *Analytical Chemistry* **2025**, *97*, 15736–15742.
- [6] Sridharan, B.; Mehta, S.; Pathak, Y.; Priyakumar, U. D. *The Journal of Physical Chemistry Letters* **2022**, *13*, 4924–4933.
- [7] Alberts, M.; Zipoli, F.; Laino, T. *Digital Discovery* **2025**,
- [8] Alberts, M.; Laino, T.; Vaucher, A. C. *Communications Chemistry* **2024**, *7*, 268.
- [9] Kanakala, G. C.; Sridharan, B.; Priyakumar, U. D. *Digital Discovery* **2024**, *3*, 2417–2423.
- [10] Wu, W.; Leonardis, A.; Jiao, J.; Jiang, J.; Chen, L. *The Journal of Physical Chemistry A* **2025**, *129*, 2077–2085.
- [11] Alberts, M.; Zipoli, F.; Vaucher, A. C. *ChemRxiv* **2023**,
- [12] Pesek, M.; Juvan, A.; Jakoš, J.; Košmrlj, J.; Marolt, M.; Gazvoda, M. *Journal of chemical information and modeling* **2020**, *61*, 756–763.
- [13] Wenk, M.; Nuzillard, J.-M.; Steinbeck, C. *Molecules* **2023**, *28*, 1448.
- [14] Hu, F.; Chen, M. S.; Rotskoff, G. M.; Kanan, M. W.; Markland, T. E. *ACS Central Science* **2024**,
- [15] Priessner, M.; Lewis, R.; Janet, J. P.; Lemurell, I.; Johansson, M.; Goodman, J.; Tomberg, A. *ChemRxiv* 10.26434/chemrxiv-2024-zmmnw **2024**,
- [16] Snyder, D. A.; Zhang, F.; Robinette, S. L.; Bruschiweiler-Li, L.; Bruschweiler, R. *The Journal of Chemical Physics* **2008**, *128*, 052313.
- [17] Szymanski, N. J.; Bartel, C. J.; Zeng, Y.; Tu, Q.; Ceder, G. *Chemistry of Materials* **2021**, *33*, 4204–4215.